# Music Connects Us: Development of a Music-Based Group Activity Intervention to Engage People Living with Dementia and Address Loneliness

**DOI:** 10.3390/healthcare9050570

**Published:** 2021-05-12

**Authors:** Hannah M. O’Rourke, Tammy Hopper, Lee Bartel, Mandy Archibald, Matthias Hoben, Jennifer Swindle, Danielle Thibault, Tynisha Whynot

**Affiliations:** 1Faculty of Nursing, University of Alberta, Edmonton, AB T6G 1C9, Canada; thopper@ualberta.ca (T.H.); mhoben@ualberta.ca (M.H.); jswindle@ualberta.ca (J.S.); dksulliv@ualberta.ca (D.T.); twhynot@ualberta.ca (T.W.); 2Faculty of Music, University of Toronto, Toronto, ON M5S 2C5, Canada; lbartel@chass.utoronto.ca; 3College of Nursing, University of Manitoba, Winnipeg, MB R3T 2N2, Canada; Mandy.Archibald@umanitoba.ca

**Keywords:** group activity, intervention design, dementia, meaningful activity, engagement, loneliness, music

## Abstract

There is a need for intervention research to understand how music-based group activities foster engagement in social interactions and relationship-building among care home residents living with moderate to severe dementia. The purpose of this conceptual paper is to describe the design of ‘Music Connects Us’, a music-based group activity intervention. Music Connects Us primarily aims to promote social connectedness and quality of life among care home residents living with moderate to severe dementia through engagement in music-making, supporting positive social interactions to develop intimate connections with others. To develop Music Connects Us, we adapted the ‘Music for Life’ program offered by Wigmore Hall in the United Kingdom, applying an intervention mapping framework and principles of engaged scholarship. This paper describes in detail the Music Connects Us program, our adaptation approach, and key adaptations made, which included: framing the project to focus on the engagement of the person living with dementia to ameliorate loneliness; inclusion of student and other community-based musicians; reduced requirements for care staff participation; and the development of a detailed musician training approach to prepare musicians to deliver the program in Canada. Description of the development, features, and rationale for Music Connects Us will support its replication in future research aimed to tests its effects and its use in clinical practice.

## 1. Introduction

Loneliness, lacking companionship or a sense of belonging [1], is a common, longstanding problem experienced by people living with dementia in congregate care homes [2]. People living with dementia in care homes are at high risk for loneliness due to interrelated, compounding factors including decreased social contact [3], communication impairments [4], death of loved ones [2,5,6], and forgetting recent interactions with significant others [7]. Loneliness predicts depression [8,9,10], increases the risk for negative emotions like sadness, fear, shame, and guilt [11,12], and contributes to responsive behaviors such as agitation and aggression [13]. Clinical interventions that target modifiable influencing factors, and which are designed specifically for use with people living with dementia, are urgently needed to address loneliness and promote the quality of life in care homes [14]. In response to this need, we describe how we applied an intervention mapping framework and principles of engaged scholarship to adapt the longstanding ‘Music for Life’ program for use in Canada. Our program, called ‘Music Connects Us’, was designed specifically to address loneliness, depression, and responsive behaviors and promote social connectedness and quality of life among people living with dementia in care homes. This conceptual paper focuses on describing the development and features of Music Connects Us, to form the foundation for future research aimed to test its effects on outcomes.

Articulating the meaning of loneliness and its antecedents is a first step in the development of an intervention to address this problem [15]. Theory can help in this regard. In the interactionist theory, championed by Weiss [1], loneliness is viewed as a normal, human reaction resulting from interrelated experiences of a lack of intimacy and poor social integration [16]. Weiss emphasized the importance of external, situational factors in producing loneliness [17]. Empirical research supports this theory and has found that loneliness is affected by situational factors like opportunities for social contact and social participation [18]. Further, personal factors like gender and marital status also derive their meaning from how individuals make sense of their socioenvironmental contexts, influencing the prevalence of loneliness and moderating intervention effects [19,20,21]. Thus, any intervention to address loneliness should be designed to target modifiable situational factors and tailored to align with people’s personal characteristics, preferences, and responses. 

Group activity interventions are multi-component, involving engagement with both an activity and with others [22], and have been used with cognitively intact older adults to address loneliness by influencing the modifiable factors of social contact [23,24,25,26,27,28,29] and social participation [23,24,25,26,27,30,31,32,33]. Developing a sense of belonging within a group setting offers acceptance and feelings of worth; intimate connections may also be formed [34]. While there is substantial literature exploring the use of group-based interventions for use with people living with dementia (using music, exercise, or cognitive training, for example), the focus of previous research has not been on loneliness, but rather on other behavioral, psychological, and cognitive outcomes [35,36]. Activity and social engagement during sessions, and the subsequent impact of engagement upon loneliness experienced by people living with dementia who are involved in group activities, remain poorly understood [35,36]. A promising avenue for future research, music-based group activities in particular show promise to engage people living with moderate to severe dementia in music-making activities and in human interactions that promote feelings of connection and wellbeing, preventing loneliness [37]. 

People living with dementia respond positively to and enjoy music, even when verbal communication is severely impaired [38]. Critically, musical function may remain better preserved in dementia as compared to verbal function and has been used as a way to communicate or connect with the person living with dementia [38]. Neuropsychological research supports that non-verbal interactions can actually have a stronger effect on promoting feelings of relating to and connecting to others than verbal interactions, because non-verbal interactions engage the right side of the brain, responsible for understanding others’ perspectives [39]. This suggests that even when people living with dementia lose their ability to speak, there remains excellent opportunity to promote feelings of connection and belonging. Furthermore, systematic reviews consistently support music as a non-pharmacologic intervention that can reduce responsive behaviors, like aggression and anxiety [40,41,42,43,44,45], which have been described as closely linked to feelings of loneliness [46]. 

Music for Life, led by Wigmore Hall, United Kingdom (UK) since 2006, is an interactive music-based program developed for use with professional musicians, care staff, and people living with dementia in care homes. In 1993, Linda Rose created the program to offer people living in a dementia-friendly society with access to exceptional musical experiences. Principles of creativity, collaboration, and equality underpin Music for Life [37]. Case studies in the UK suggest that Music for Life can promote feelings of connection for people living with dementia based on their positive verbal and non-verbal responses noted by project leaders during sessions [47]. Research about the program to date includes an ethnographic study focused primarily on the experiences of musicians during sessions [37], and a recent (2020) grounded theory to identify ways in which eight people living with advanced dementia communicated and interacted during Music for Life sessions [48]. This latter study identified verbal and non-verbal communication during sessions, including talking, smiling, laughing, or using dance-like movements, and social interactions such as turn-taking, humor, and mirroring one another’s actions [48]. The findings highlight the promise of interactive music sessions to engage people living with dementia in music-making and social interactions with others; further research is needed to assess the extent and character of engagement behaviors over the course of these 60-min interactive music sessions to inform future intervention research aimed to rigorously test program impact. 

Our team has adapted this promising music-based group activity in preparation for future evaluation studies when it is used as a complex intervention in Alberta, Canada. Our adapted program is called Music Connects Us. Assessing Music Connects Us in future research will advance our understanding of engagement behaviors, and ultimately aim to address the pervasive loneliness of care home residents living with dementia. The purpose of this foundational conceptual paper is to describe the approach we used to adapt the Music for Life program for use in Alberta, Canada, outline the features of our adapted program to support its replication in future research and practice, justify the most important program adaptations, and highlight key directions for further research. 

## 2. Approach to Adaptation of Music for Life to Design Music Connects Us

In our previous research, we applied scoping review and mixed methods methodologies to identify the elements of interventions that target loneliness experienced by people living with dementia (i.e., intervention mapping steps 1 and 2). The present manuscript reports upon the collaborative, conceptual work that we undertook to complete intervention mapping step 3: the organization of the strategies identified in our previous research into a program for use in the target population. To complete step 3 of intervention mapping, our team worked collaboratively to describe the elements and activities involved in Music Connects Us and to explicate its theoretical underpinnings, reflecting principles of engaged scholarship. Intervention mapping [15] and engaged scholarship [49,50], the overarching approaches which underpinned our adaptation work, are described in more detail below. 

Our process to develop Music Connects Us can be understood within an intervention mapping framework. Intervention mapping involves a review of the literature to identify empirical findings, the identification and application of relevant theory, and collecting new data [15]. Intervention mapping proceeds in several steps [15]. Step 1 is focused on identifying the program goal and specifying who and what will be changed by the intervention in the short-term, while Step 2 is focused on the identification of theoretical approaches and practical strategies that will achieve the proximal program objectives. To develop Music Connects Us, steps 1 and 2 involved a scoping review to define the targets, components, activities, and modes of delivery of group activities that address loneliness [22], and a mixed methods study to assess stakeholders’ views related to the use of group activities for use with people living with dementia (manuscript in preparation). Steps 4 and 5 of intervention mapping are focused on implementation and planning for the process and effect evaluation of the program and will be addressed in future studies. This manuscript focuses on Step 3 of intervention mapping, a more conceptual phase in which we organized the strategies to address loneliness into a program for use with the target population, and produced program materials. 

We applied principles of engaged scholarship [49,50] to design Music Connects Us and complete Step 3 of intervention mapping. Our community musician partners approached us with the idea to adapt Music for Life for use in Canada, and we worked closely with these community partners to develop Music Connects Us. The academic team brought to this project expertise related to loneliness, intervention design, research with people living with dementia and care home residents, music and arts-based interventions, and mixed methods evaluation design. Our community partner, the Winspear Centre, brought to the project musical expertise as a Canadian symphony organization that aims to implement an interactive musical program with people living with dementia in Canadian care homes. Our international partner, Wigmore Hall, brought expertise with the use of Music for Life, the program that applies musical improvisation to engage people living with dementia in care homes in the UK. Music for Life is not a trademarked or franchised program, and Wigmore Hall is enthusiastic about adaptation of the program for use in Canada. Collaborators from continuing care organizations and the Government of Alberta brought expertise in the care of people living with dementia in congregate care settings, and a commitment to both pilot and use the program, once found effective, to improve quality of life of care home residents. Our interdisciplinary, international team shares the vision of evaluating the processes and impacts of Music Connects Us and identifying ways to optimize is spread and sustainability within Canada. To work in a collaborative way with our team, we held planning meetings with community partners and collaborators, reviewed Music for Life materials, and consulted with musicians.

### 2.1. Planning Meetings

Our team held three days of face-to-face planning meetings at the University of Alberta, Canada, in June 2018. The overall goal of the meetings was to discuss our team’s planned approach to adapt the Music for Life program, considering feasibility for use in Alberta, Canada. These meetings involved research and program presentations, brainstorming and discussion sessions, and site visits. Attendees for all three full days included two members of the Winspear executive, four researchers with expertise in the arts, music, elder care and intervention design, the Music for Life program manager from Wigmore Hall, and the executive director for the Institute for Continuing Care Education and Research (ICCER). On day two, Edmonton Symphony Orchestra musicians, an Alberta continuing care policymaker, and five dementia care service providers who expressed interest to serve as pilot sites also joined the meetings to meet the team, review research and program presentations, and to raise issues and discuss their questions. Notes from the presentations and discussions during the meetings identified the key features of an adapted intervention, rationale for adaptations, and the research questions for a future pilot study.

#### 2.1.1. Research and Program Presentations 

On days 1 and 2, research and program presentations aimed to orient the group to the science underpinning the future pilot study and show the synergistic value of the team members’ roles and expertise. Presentations related to group activity interventions (O’Rourke), music interventions (Bartel), and the Music for Life program (Wigmore Hall representative). ICCER’s executive director described the organization’s role in supporting continuing care organizations’ involvement in this research and identifying opportunities to disseminate research findings within the continuing care sector in support of practice change. Presentations were delivered on day 1 to the group of researchers, and the delegates from Wigmore Hall, the Winspear and ICCER. On day 2, versions of these presentations appropriate for the audience of Edmonton Symphony Orchestra musicians and continuing care policymaker and service providers were delivered to promote further discussion. Brainstorming and discussion sessions, facilitated by O’Rourke, then followed the didactic presentations. 

#### 2.1.2. Brainstorming and Discussion Sessions 

Days 1, 2, and 3 included brainstorming and discussion sessions to: (i) identify the key research questions of interest to the group; (ii) discuss the opportunities and challenges that this group may experience, and strategies to mitigate challenges in order to build a cohesive, interdisciplinary, international team; (iii) consider issues related to intellectual property and fees for service, and begin to develop a framework for meeting the group members’ needs in this regard; and (iv) consider the similarities and differences between the UK and Alberta, Canada contexts to discuss specific program adaptations. 

#### 2.1.3. Site Visits 

On day 3, the researchers, ICCER director, and Winspear and Wigmore Hall partners visited several congregate care sites that expressed interest in participating in the Music Connects Us program. In Alberta, congregate care includes facility living (long-term care) and several levels of supportive living (SL) [51]. For this project, we will be working with SL 4 and/or 4D (includes people with advanced care needs living with dementia), and long-term care facility residents. The intent of these site visits was for all members of the team to develop an understanding of the context for where the intervention would be implemented. These visits further informed discussions on Day 3 as to how the Music for Life program may need to be adapted for use in Alberta, Canada. 

### 2.2. Consultation with Musician Team Members 

In the months following the planning meetings, we developed a detailed description of the Music Connects Us program and a manual to describe musician training to deliver Music Connects Us. We developed these materials based on review and synthesis of our planning meeting notes and available Music for Life program documents, including a book describing features of the program from the perspective of musician participants [37] and unpublished documents provided to us by our Wigmore Hall partners (e.g., presentation slides). We then consulted with some of the musician members of our team by phone to address remaining questions. We provided the consultants with our questions, and summaries of Music Connects Us and the proposed musician training program, in advance of the phone call. All consultants were provided an honorarium for their time. We consulted with three program directors and musicians involved in the UK Music for Life program to add detail to the description of Music Connects Us and our proposed training approach, particularly in relation to the characteristics of musicians, and how they were selected and trained to deliver their program. We consulted with five musicians or program leaders from the Winspear Centre’s Edmonton Symphony Orchestra (ESO) to ask them questions about the clarity, logic and usefulness of the training program which we had developed. We revised the description and summary of the training program to clarify it based on their feedback. 

## 3. Products of the Adaptation Process

In this section, we will describe the main product of the adaptation process, the description of the Music Connects Us program, followed by the key adaptations that were considered in developing this program to fit the context in Alberta, Canada.

### 3.1. Description of Our Adapted Program: Music Connects Us

#### 3.1.1. Goal and Components

Music Connects Us is a music-based group activity intervention to promote engagement and address loneliness and related outcomes for use with older adults living with moderate and severe dementia residing in care home settings in Canada. The primary components of this intervention target modifiable factors which influence loneliness, including social contact and social participation. The two mechanisms through which these factors are targeted are engagement in an activity (i.e., music-making) and in interactions with others. Moderators are factors that may influence the impact of the program on outcomes and include gender and the level and nature of engagement during sessions (Figure 1). 

#### 3.1.2. Who Delivers Each Project?

Responsibility for the project management is shared between the research project leads who organize sites, oversee training, and supervise research assistants and program delivery, and a Winspear Centre partner. The Winspear Centre partner supports musician scheduling and selection (e.g., from the Edmonton Symphony Orchestra and the Youth Orchestra of Northern Albertan Teaching Artists), and locates appropriate space for musician training. Professional and semi-professional musicians are recruited and are asked to identify their students that may be a good fit for the program. A facilitator supports the musicians to deliver sessions. All are asked to commit to a growth mindset, a philosophy of person-centred dementia care, self-reflection and mindfulness, and clear and open communication throughout the project. 

#### 3.1.3. Session Participants and Activities

For each Music Connects Us project, a consistent group of eight men and women living with moderate to severe dementia participate in Music Connects Us sessions once per week for eight weeks. Residents living with dementia are selected prior to the implementation of the project. The project manager describes the purpose of the program and inclusion criteria and asks the facility manager and care staff to identify approximately 15 residents over 65 years of age who are diagnosed with dementia and live with moderate to severe cognitive impairment. Residents who express poor communication, display responsive behaviors, or are frequently withdrawn should be specifically considered.

A selection meeting is then held between the facility manager, the lead musician, the facilitator, and the project manager to create a ranked list of the 15 residents who will be invited to partake in Music Connects Us. The facility manager (or other care staff) approaches these individuals or their designated decision makers in sequence to ascertain their interest, and the research assistant then obtains consent to participate from interested residents, until eight residents have consented. If someone drops out, another resident can be approached to take their place. The facility manager identifies a quiet activity space of the care home for the sessions. The space should minimize distractions, promote a feeling of intimacy, and comfortably seat all participants and musicians in a circle.

Prior to each session, the musicians and facilitator arrange seating in a circle, welcome residents to the session, and support them to become familiar with their physical space. Consistent assigned seating promotes familiarity. Each session lasts one hour, to allow time for individualized engagement of all group members, without causing fatigue. The session begins by offering each resident an instrument to use. This is an opportunity for residents to express their individuality and autonomy. Sessions open (and end) with a framing composition, which the musicians create over about 2 h together in advance of the project. The framing composition is a four-bar instrumental fragment and used as a motif to begin (and end) each session. Improvisation upon this motif does occur (i.e., changing the character, speed, and dynamics). For example, at the end of a very active session, if may be delivered at a slower, more calming tempo. A welcome song follows the framing composition. The name of each resident is sung, led by one musician (e.g., the designated lead musician or whoever is sitting closest to the residents), while a base instrument plays to ground the song. Musicians discuss the welcome song before the project, or they may choose an existing welcome song that has been used for other projects. 

The main Music Connects Us activities begin after the welcome song, when focus shifts to creating musical pieces focused on each participant, validating the personhood and experience of each resident. Residents are assessed throughout the session and approached at least once in a session to ascertain their interest (based on non-verbal and verbal cues) in being the focus of an improvised piece. Residents can be approached more than once during the session. Focused pieces last no more than a few minutes and are led by one musician and accompanied by the other musicians. All eight residents engage in any way they desire: actively observing, playing their instrument, moving their bodies to the music, or interacting verbally or non-verbally with others in the group. Once the piece is finished, the musicians choose another participant and begin another focused piece. 

When a focused piece is being created, one or more of the musicians move closer to that resident and respond to the musical (e.g., tapping a drum, humming, tapping one’s leg or foot) and non-musical emotional cues (tears, smile, outstretched arms) to create music with the resident, and which reflects how the resident is feeling and responding. Musical frameworks and a range of musical and improvisational techniques are used to motivate interactions with individuals and between the group, such as repetition, imitation, and matching (Table 1). Even noises (e.g., a repetitive vocalization or foot tapping) can be incorporated into or used to inspire a piece of music, meeting the resident where they are in that moment, connecting with them, and including them to create music together. Music that is familiar to or sang by a resident in that moment can be used to guide musical motifs, which are then varied or transitioned to other elements to work creatively with residents. Musicians communicate with individual residents with personalized compositions but maintain an awareness of everyone else in the group. After 60 min, the prepared closing song is played.

Following each session, the facilitator leads the musicians in a 30-min reflective debrief discussion. During the reflective debrief, the facilitator guides the musicians to consider their musical and relational experiences during that session, and identify challenges and strategies related to engaging residents during the sessions. The facilitator comments on the observations they made of musicians and residents during the session and may offer recommendations for future sessions. Musicians can also take the lead by sharing their experiences. This discussion is held in a separate room and prompting questions such as “did you see anything that surprised you?” and “did you learn anything new about a resident or colleague?”, are used to facilitate the debrief. The debrief is an opportunity for the group to develop a sense of cohesion and for the facilitator to support musicians. The opportunity to reflect and debrief as a group helps to build musicians’ self-awareness, resilience, and empathy. 

#### 3.1.4. Characteristics of the Facilitator

The facilitator is a research assistant who has expertise in person-centered dementia care and a background in music, to allow them to communicate effectively with the musicians. For example, a recreation therapist or registered nurse with formal musical training, who has worked closely with people living with dementia in care homes, would be ideal for this role. The facilitator helps to set up each session, attends each session to observe and take field notes related to the tone of the session and their observations of engagement and the challenges or strategies used to engage participants. If engagement is to be formally assessed, then the facilitator also sets up a 360-degree video camera to record the sessions. Engagement can be rated by the facilitator or another research assistant following sessions using the Observed Emotion Rating Scale to quantify the extent and nature of resident engagement during the session (measured with the interest subscale, and positive and negative affective state subscales) [52]. The facilitator should demonstrate empathy, direct communication, diplomacy, and sensitivity.

#### 3.1.5. Musician Characteristics

Three highly trained musicians lead the sessions (2 professional or semi-professional and 1 advanced student). The aim is to have a consistent musician trio deliver all 8 sessions to a group of residents, and the musicians’ styles and personalities should be complementary. Each musician trio plays a different instrument or sings to offer a range in musical tone responsive to different moods and preferences. Musicians’ backgrounds (e.g., symphony orchestra, chamber orchestra, contemporary) and specialties (e.g., bass, melody line, countermelody) may vary. Musicians begin with a high level of technical skill (i.e., pitch, ability to play without having to read music), and develop improvisational proficiency during training and program delivery. One musician leader in each group is designated to develop and maintain a supportive environment among the three musicians. They require leadership skills to guide the flow of the sessions, create trust and confidence within the musician group, encourage reflection, and reliably develop musical elements, and themes. The musicians’ instruments are selected to work well within the circle of participants and should not overpower the session. This is one reason why pianos are not typically used by musicians. 

Musicians are patient, flexible, self-reflective, and open to showing vulnerability. The musicians use language that is genuine and affirming, ensuring to be gentle and clear, and use open facial expressions and touch such as hand-holding to help build a rapport with the residents and make them feel more comfortable in the session. Musicians consider both space and pace throughout sessions. Pace is giving an individual enough time to process an interaction that has taken place. Space is listening openly to participants and creating an environment where authentic interactions can occur. Space can refer to physical space, musical silence, or stillness in body language.

#### 3.1.6. Selecting Instruments for Residents

Instruments are selected which can stimulate the senses, have interesting tactile characteristics, and should be easy to use (but not childish). Budget constraints must also be considered, as well as how easy they are to transport and clean. Instruments are displayed before the start of the session in a way that will be appealing to participants. Examples of instruments used by residents may include the ghungroos, djembe, tingsha bells, shaker, caxixi, maracas, metallophone bass bar, soprano bass bar, hand chime, kalimba, spirit chime, agogo bell, bodhran, cabasa, castanets, claves, gato drum, guiro, frame drum, ocean drum, rainstick, tambourine, and wood block. Residents may also choose not to use an instrument, but may use the voice, clap, or tap their feet, for example.

#### 3.1.7. Activities to Support Sustainability

Care staff who attend and reflect upon Music Connects Us sessions may see the person with dementia in a different way and may continue to incorporate music into residents’ lives after the project ends. We will invite three care staff to each session to explore how including them may help sustain the positive effects of Music Connects Us after the eight-week program ends. We will aim to include the same care staff within each group, whenever possible. Care staff who attend each session can play an instrument or simply observe. They may speak and interact with participants during the sessions but will be asked not to push any person living with dementia to engage in any way that they do not want in that moment. Care staff could be from any professional background, so long as their work involves the care and wellbeing of persons living with dementia. Examples of care staff may include care aides, registered nurses, unit managers, or recreation therapists. 

### 3.2. Summary of Adaptations Applied to Create Music Connects Us

#### 3.2.1. Program Conceptualization and Framing

The first main consideration for adaptation was related to conceptualization of the primary program aim, an essential piece in explicating the theory of the intervention (i.e., the elements of Music Connects Us and how it is proposed to affect loneliness) [53]. In Music Connects Us, emphasis is on the experiences of the person living with dementia, and exploration of musician and care staff experiences is secondary to this primary aim. Conversely, Music for Life consists of three interlinked, and equally weighted, strands of work: (1) musical improvisation for the residents with dementia to connect and communicate in new ways, validate their experiences and identity, and build new relationships; (2) experiential staff training, to deepen their understandings of the residents, promote person-centred care and teamwork, and increase their confidence in their work; and (3) training for professional musicians to work with people living with dementia, create relationships with the residents, and increase their musical, creative and leadership abilities. We discussed how each strand was important to the program but could be re-framed in our adapted program. Our team wanted to focus on the experiences of care home residents living with dementia in Music Connects Us sessions, addressing the knowledge gap related to use of music-based group activities to promote engagement and address loneliness. Music Connects Us is focused on how to target factors of social contact and social participation to address loneliness experienced specifically by people living with dementia, and this is reflected in the stated intervention goal, targets, and outcomes (Figure 1). This approach fit with the research expertise of the study lead and aligned with the interests of our continuing care and decision-maker partners to understand how the program impacted people living with dementia. 

#### 3.2.2. Reflective De-Brief Participants

The second main consideration for adaptation was related to who would participate in the reflective debrief session. In Music for Life, the debrief is aimed to support both the musicians and the care staff to develop a reflective practice [54]. In contrast, in Music Connects Us, de-brief sessions are intended primarily as ongoing support for the musicians to deliver high quality Music Connects Us sessions. To decrease program costs and enhance feasibility, we include only musicians in the de-brief session, and limit the de-brief to 30 min (as compared to Music for Life’s 60-min debrief for 3 musicians and 3 to 5 care staff). Professional and semi-professional musicians are each paid around $75 to $150 CA per hour, so the length of the de-brief affects program costs. In addition, our continuing care provider partners confirmed that care staff time is a limited resource, so we would need a strong justification to include care staff in the de-brief. In future studies, we will record the de-brief sessions with musicians, and invite care staff to participate in a focus group at the end of the program, to gather data related to their experiences, perceptions of impacts of the program, and reflections related to sustainability of program outcomes to inform future decisions related to their inclusion in the de-brief.

#### 3.2.3. Program Offerings per Care Home

The third main consideration for adaptation related to the number of times that the program is offered within a care home per year. Music for Life sets the number of residents to eight per session and delivers sessions to just one group of eight in a care home at any one time. Following our tours of the care homes, our Wigmore Hall partners remarked upon the similarities of the care home context and populations to the settings where they deliver the program in the UK. However, a key difference was noted in the size of the care homes: they had not delivered the Music for Life program in a care home with such a large resident population, yet homes with 100 or more residents are common within Canada. Given this, our partners discussed how we may have a difficult time choosing just eight participants from a single, large care home. As a result, we will propose to offer Music Connects Us to two groups over one year within each care home.

#### 3.2.4. Composition of Musician Group

The fourth main adaption of the program was related to the composition of the group of three musicians who would be selected to deliver each session. Again, this discussion was prompted by a concern for the feasibility and sustainability of the program within Canada. The planning meetings identified several approaches to reduce the costs of each session including involving music students and other semi-professional community musicians. While our Edmonton Symphony Orchestra partners are enthusiastic about spearheading the development and implementation of Music Connects Us, professional symphony musicians have very busy rehearsal and performance schedules, which limits their availability to participate in programs like Music Connects Us. Involving student and other community musicians will allow us to train more musicians and increase the number of offerings of Music Connects Us per year. This addresses our continuing care partners’ and Government of Alberta requests to offer the program more often within continuing care sites once its effectiveness is established following a future pragmatic clinical trial. 

#### 3.2.5. Development of Training Program

The fifth main adaption to the program was the development of an explicit training program for musicians. We discussed at length the importance of training musicians to engage with people living with dementia and to improvise musically. However, the complex task of training Music for Life musicians had not previously been described in written materials. Our discussions and consultations with Wigmore Hall highlighted that their current training approach relies primarily upon an apprenticeship model where new musicians observe and participate within projects lead by experienced Music for Life musicians. Based on our planning meetings, and our consultation with Winspear and Wigmore hall musicians and program leaders, we have developed a three-phase musician training approach that involves a 28-h program to: (1) learn about person-centered dementia care, (2) practice musical improvisation, and (3) observe and practice in sessions lead by an experienced Music Connects Us musician (or a Music for Life musician in the first iterations of the Canadian program). Phase 1 is completed by Canadian musicians in advance. In the first iteration of the program in Canada, Phases 2 and 3 will occur with the support of in-person Music for Life musicians during a 1-week intensive. 

Phase 1, learning about person-centered dementia care (9 h total), involves review of the detailed training manual (3 h), completion of online modules and review of several additional resources (3 h), and a visit to a long-term care home (2 to 3 h). Knowledge of person-centered care, effective communication skills, and strategies to help interpret the person with dementia’s actions and expressions are essential to support the musician to engage the person living with dementia through music. Phase 2, musical improvisation (6–7 h total), involves an introduction to Music Connects Us (1 h), an improvisation half-day workshop led by a Music for Life musician (3 h), and an improvisation half-day to allow practice between Canadian musicians (2–3 h). This phase helps to refine a musician’s skills in improvisation. Phase 3, Music Connects Us sessions (12 h total), involves observation and participation in eight mentored Music Connects Us sessions over three weeks. Each session lasts 1 h, and is followed by a 30-min de-brief period to discuss in detail the musicians’ experiences of the session. Sessions one to six occur over four days during the 1-week training intensive. Music Connects Us trainee musicians start in a mostly observational role, but by sessions five and six, each of the trainee Music Connects Us musicians will have participated in two sessions (and have observed other sessions) while the Music for Life musicians observe and provide a supportive role. Sessions seven and eight occur in the two weeks following the training intensive. These sessions are led by the trainee Music Connects Us musicians. Video footage is shared with experienced Music for Life musicians, who will offer expert consultation by distance related to the quality of the program delivery. 

## 4. Discussion

We propose that theory-driven, music-based group activity interventions like Music Connects Us can be used specifically to engage people living with dementia in order to promote feelings of social connectedness and increase quality of life. In this conceptual paper to describe the features of the Music Connects Us program, we have detailed the development of Music Connects Us and outlined the collaborative process that we applied to adapt a previous program, while mapping the intervention components and activities to empirical literature and theory. People living with dementia in care homes require a richly textured living environment to support their quality of life [55], and music has a clear role to play in this agenda. Music Connects Us involves a range of professional, semi-professional and student musicians who create musical experiences, and is part of a broader movement to increase use of music in care environments. This broader movement involves a range of initiatives including: music therapy, where trained therapists use music prescriptively for health and wellbeing; music medicine, where the sonic, vibrational, and rhythmic properties of music and sound are used as medicine; other kinds of music-based programming, like offering care home residents personalized playlists; and organizations like the Room 217 foundation which offers caregivers, with and without musical skill, the training and resources to integrate music into many aspects of their practice in order to deliver ‘music care’ [56]. Music Connects Us primarily addresses the music care domain of musicking, which is the promotion of spontaneous and informal engagement with music (e.g., listening, music-making) [57], and offers residents a specialized, time-limited musical experience. The time-limited nature of Music Connects Us, as well as the fact that it can only reach select residents from a care home, are key limitations to its potential impact. Another concern related to the program is that the use of community musicians requires buy-in from a large number of stakeholders in order to spread and scale-up the program, as well as commitment from organizations to pay musicians after research funding ends. Our team raised these concerns, and we continue to explore how to address these issues moving forward. Including care providers in sessions may prime care home organizations to become interested in a broader ‘music care’ program such as that offered by the Room 217 Foundation and which aims to engage all care providers to integrate music into residents’ daily lives [57], bringing complementary interventions together to extend their positive effects. Future research should explore the complementary, synergistic qualities of distinct music-based interventions with the aim to develop the evidence base for various uses of music in care environments. Future research is also needed to assess the costs and benefits of Music Connects Us in relation to other music-based approaches, to determine how far this innovative, yet potentially costly, program can and should be spread in Canada.

Intervention mapping [15,58] and the Medical Research Guidance for the development of complex interventions [59,60] highlight the importance of undertaking extensive conceptual work to develop an intervention. Intervention conceptualization involves review of theory, empirical literature, and conducting empirical studies to ensure that the problem and its antecedents are well-understood, and to identify possible solutions which will address the modifiable influencing factors [61]. Prior to intervention mapping, work is done to understand the problem and its antecedents. Our work prior to developing Music Connects Us included completing a meta-synthesis to understand the factors that affect quality of life from the perspectives of people living with dementia which underscored the importance of social connectedness [14], and a scoping review to define and identify the antecedents, indicators, and outcomes of social connectedness for older adults [18]. These previous studies were essential to determining intervention mechanisms so that how the intervention is hypothesized to influence loneliness and quality of life in the target population could be clearly explicated [59]. Once the problem and its antecedents are well-understood, intervention mapping can be then used to develop health promotion interventions [15,58,61]. 

Previously, descriptions of interventions developed for older adults to address loneliness have been sparse, identifying the name of the activity and including the mode and frequency of delivery, but the specific activities and theoretical underpinnings were not described in the detail required for replication studies to test the interventions’ mechanisms [22]. Recent recommendations call for clear and detailed descriptions of the theory of interventions in conceptual papers to allow for adequate description of their goals, components, mechanisms of action, and the factors that influence intervention delivery [53]. In describing the components of a music-based group activity intervention, this paper contributes to the development and conceptualization of health interventions, supporting its replication for use in future research and clinical practice [53].

By applying principles of engaged scholarship, we are confident that key stakeholders including policy makers and continuing care providers, view the problem of loneliness as essential to address in order to improve the quality of life of people living with dementia. The core principles of engaged scholarship include high quality scholarship, reciprocity, identified community needs, knowledge democratization, and boundary crossing [62]. The program of work to develop Music Connects Us has demonstrated a strong commitment to identification of community needs, starting with determining the value of social connectedness from the perspectives of people living with dementia [14] and elicitation of perspectives of family, friends and health care providers related to the use of group activity interventions with people living with dementia (manuscript in preparation). However, missing from our meetings to specifically discuss Music Connects Us were people living with dementia and their family and friends. Future collaborative work to discuss this program should include representatives from these groups to ensure that the program remains relevant and addresses their concerns. 

A key strength of our work, the need for an adaptation of Music for Life for use in Canada was identified by local community musicians and received enthusiastic support from policy makers and continuing care providers within the province. By working in an interdisciplinary team and with community-based collaborators, we are crossing boundaries. However, this work is also framed in a way typical to the health sciences, with a primary focus on health and quality of life outcomes of people living with dementia. While this was the most appropriate framing for this project at this time, future research to explore this program from different angles could apply a more social science or arts-based lens to design studies about Music Connects Us which prioritize the experiences of musicians or continuing care staff. 

This work is a first step in a larger program of research and remains closely tied to traditional academic structures. Our meetings and discussion sessions were focused on the design of a pilot study, lead by a University of Alberta faculty member, and for implementation in Alberta, and included stakeholders essential to this specific project; however, future research aimed to spread Music Connects Us must also include stakeholders from other jurisdictions and continue to explore the most meaningful ways to engage with those stakeholders (e.g., hosting a World Café). To this end, future sessions with larger groups of stakeholders would use small group activities and be facilitated by someone other than a research lead, in order to ensure all voices are heard and to further break down any hierarchical structures which exist in the team. 

Our collaborative work to develop Music Connects Us highlights several gaps in knowledge that we can address during future pilot and effectiveness testing of the program during steps 4 and 5 of intervention mapping including: What is the extent and nature of engagement behaviors and mood displayed during Music Connects Us sessions by men and women living with moderate to severe dementia in care homes?How do engagement behaviors change over the course of a 1-h session?What are the effects of musical engagement on people living with dementia? For example, can participating in a weekly session improve feelings of loneliness, and related outcomes like responsive behaviors, depression, and quality of life?

The next steps in this program of research are to conduct an in-depth exploration of the processes of this music-based group-activity intervention, and advance understanding of engagement behaviors during sessions for people living with dementia. We will continue to work closely with our Wigmore Hall partners to ensure quality delivery of our program and to share materials and learnings from each stage of the evaluation. Moving in this stepwise approach, we are developing the foundation for further research and practice to activate engagement among people living with dementia in care homes. This novel program of research will, ultimately, test how music-based group activity engagement can address loneliness and quality of life in an understudied population and aims to contribute to the diverse and growing body of research aimed to promote the use of music in healthcare.

## Figures and Tables

**Figure 1 healthcare-09-00570-f001:**
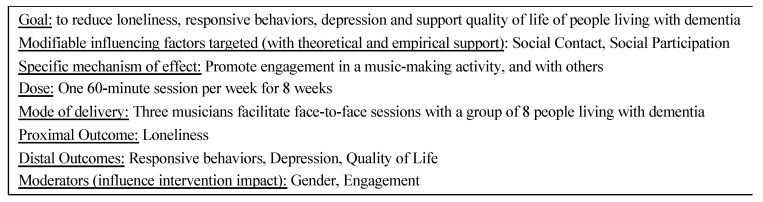
Summary of the Music Connects Us intervention theory including its goals, components, activities, mode of delivery and dose.

**Table 1 healthcare-09-00570-t001:** Musical techniques for use in Music Connects Us sessions.

Technique	Description
Repetition	Repeating a sound, tone, melodic phrase or other musical material; involves improvising on a familiar musical style and genre to emphasize a feeling or idea to create rhythm.
Scaffolding	Adding new elements to repeated musical material to progress an improvised piece.
Modeling	Giving a clear example to follow, and most commonly used when introducing a new instrument.
Imitation	Where the musician mimics an exact copy of the resident’s presentation.
Mirroring	Copying the music that the resident plays and their body language. Used to encourage the resident to continue or expand upon their musical motif and promote empathetic connection.
Matching	Emulating the style and quality of music that a resident has played, to build upon what the resident has played in a congruous way.
Reflecting	Creating music that reflects the resident’s mood or underlying communication, as read by the musician; used to promote empathetic connection.
Translation	Playing music that validates what the resident has done, and then linking this to another type of musical contribution.
Hammer	A type of translation that creates a new energy and group dynamic.
Silence	Absence of sound and can be used before and after improvisations. It is powerful and allows time to recognize the resident’s engagement.
Singing	Use of the voice can be done at any time during the session in a strong voice with a good sense of pitch to engage or respond to the residents and make personal connections.

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
