# Peer review of "Music Connects Us: Development of a Music-Based Group Activity Intervention to Engage People Living with Dementia and Address Loneliness"

_healthcare, 2021, doi:10.3390/healthcare9050570_

Round 1

Reviewer 1 Report

The article addresses an interesting question, and in this sense, there is a relevant strength, that is the used of music to help people living with dementia.   

There are some issues that, I believe, could contribute to improve the paper:

One of the strengths of the paper is that the literature review is extensive and has covered many of the key concepts related to the paper. Another strength is that the program is well describe. However, regarding the first point a concept that could be taken into account could be the motivational variables in the teaching music context (self-concept, causal attributions, content goals, and the relationships between all of them). In this sense, I recommend that in the introduction this issue could be considered. A reference that should be used is:

Holgado-Tello, F. P., Navas, L., & Marco, V. (2013). The Students' Academic Performance at the Conservatory of Music: A Structural Model from the Motivational Variables. Journal of Psychodidactics, 18(2).

Regarding the program, I miss some relevant information in order to valorate it’s efficacy and effectiveness. If it is possible, should be reported information about:

  • The sample (participants, inclusion and exclusion criteria, experimental mortality, missing rate,…)
  • Cuasiexperimental design used to probe the program.
  • Instruments used to measure the dependent variables (loneliness, responsive behaviors, depression and quality of life).
  • Control techniques used to control confounding variables.
  • Statistical analysis to compare the group or groups.
  • Results of how the program improve the score in the dependent variables.

If there aren’t information about these aspects that allow valorate the program, a clear justification about it should be given. May be the general objective of the manuscript is not clear, at least, for me.

Also, information about how the discussion sessions were systematized and which method was used could be welcoming.

In sum, the paper is well written, and it could contribute to the accumulation of empirical evidences about how the quality of life of people living with dementia could be improve thorough the music. In this sense is an interesting paper that could increase the literature with relevant results.

In conclusion, I would like to emphasize that I understand that the vision of the reviewers, at times, can seem idiosyncratic, and from this position I would not like to be dogmatic or rigid in my approaches. I have only tried to offer some suggestions on how the article could be improved, which may or may not be considered.

Reviewer 2 Report

I find this article to be of high interest for the field. There is however lots of evidence that music is (also) important for people with dementia. I really enjoyed reading such a theoretical driven article that link music and group activities. I have only three concerns that would be interesting to see the authors responses to:

  1. the structure is a littlebit confusing. What kind of methodolgy is used and why is results used. Maybe clarify a bit.
  2. some reflections on limitations of the methods should be added
  3. some more critical reflections on the use of MCU should be discussed
